# High-grade serous ovarian carcinoma organoids as models of chromosomal instability

**Maria Vias[1†], Lena Morrill Gavarró[1,2†], Carolin M Sauer[1], Deborah A Sanders[1], Anna M Piskorz[1], Dominique-Laurent Couturier[1], Stéphane Ballereau[1], Bárbara Hernando[3], Michael P Schneider[1], James Hall[1], Filipe Correia-Martins[1], Florian Markowetz[1], Geoff Macintyre[3], James D Brenton[1]***

[1]Cancer Research UK Cambridge Institute, University of Cambridge, Li Ka Shing Centre, Cambridge, United Kingdom; [2]The MRC Weatherall Institute of Molecular Medicine, Oxford, United Kingdom; [3]Centro Nacional de Investigaciones Oncológicas, C/Melchor Fernández Almagro, Madrid, Spain

**Abstract** High-grade serous ovarian carcinoma (HGSOC) is the most genomically complex cancer, characterized by ubiquitous *TP53* mutation, profound chromosomal instability, and heterogeneity. The mutational processes driving chromosomal instability in HGSOC can be distinguished by specific copy number signatures. To develop clinically relevant models of these mutational processes we derived 15 continuous HGSOC patient-derived organoids (PDOs) and characterized them using bulk transcriptomic, bulk genomic, single-cell genomic, and drug sensitivity assays. We show that HGSOC PDOs comprise communities of different clonal populations and represent models of different causes of chromosomal instability including homologous recombination deficiency, chromothripsis, tandem-duplicator phenotype, and whole genome duplication. We also show that these PDOs can be used as exploratory tools to study transcriptional effects of copy number alterations as well as compound-sensitivity tests. In summary, HGSOC PDO cultures provide validated genomic models for studies of specific mutational processes and precision therapeutics.

## Editor's evaluation

This fundamental work substantially advances our understanding of patient-derived organoids as a useful model to evaluate chromosome instability and identify novel therapeutic strategies to combat HGSOC. The study is comprehensive, and the evidence supporting the conclusions is compelling, which would further benefit the related research about the mechanisms of genomic instability in HGSOC.

***For correspondence:**
james.brenton@cruk.cam.ac.uk

[†]These authors contributed equally to this work

## Introduction

HGSOC is a heterogeneous, chromosomally unstable cancer with predominant somatic copy number alterations (SCNAs) and other structural variants including large-scale chromosomal rearrangements (*Li et al., 2020*; *Drews et al., 2022*). Oncogenic mutations are rare and recurrent somatic substitutions involve less than 10 driver genes (*Ahmed et al., 2010*; *Cancer Genome Atlas Research Network, 2011*; *Gerstung et al., 2020*; *Aaltonen et al., 2020*). Loss of p53 function and *TP53* mutation have been defined as the earliest driver events permitting the development of diverse chromosomal instability (CIN) phenotypes that are apparent in pre-invasive lesions in the fallopian tubal epithelium (*Ahmed et al., 2010*; *Labidi-Galy et al., 2017*). Mutational signatures are genomic

patterns that are the imprint of mutagenic processes accumulated over the lifetime of a cancer cell (*Petljak et al., 2019*). Genome-wide patterns of single nucleotide (SNV) (*Alexandrov et al., 2013*) and structural variants (SV) (*Davies et al., 2017*) showed that these mutational spectra could identify specific mutational processes with the majority of cancers having multiple signatures. We have previously shown that shallow whole genome sequencing (sWGS) methods can stratify HGSOC based on the distributions of six copy number features that encode patterns of different causes of CIN (*Macintyre et al., 2018*; *Cheng et al., 2022*). These copy number signatures are able to recapitulate the major defining elements of HGSOC genomes, including defective homologous recombination (*Cancer Genome Atlas Research Network, 2011*), *CCNE1* amplification (*Etemadmoghadam et al., 2009*), amplification-associated fold-back inversions (*Wang et al., 2017*) and are associated with distinct tumor-immune microenvironments (*Jiménez-Sánchez et al., 2019*). These methods have also been applied to molecular stratification of testicular germ cell tumors and multiple myeloma (*Loveday et al., 2020*; *Maclachlan et al., 2021*). This work provides the first generalized classifier for HGSOC that focuses on the mechanistic basis of CIN (*Drews et al., 2022*; *Macintyre et al., 2018*).

CIN drives clinically relevant genetic and cellular phenotypes including extrachromosomal DNA and micronuclei (*Turner et al., 2017*; *Zhang et al., 2015*), activation of innate immune signalling (*Bakhoum and Cantley, 2018*), metastasis (*Bakhoum et al., 2018*; *Turajlic et al., 2018*), and therapeutic resistance (*Ippolito et al., 2021*; *Lukow et al., 2021*). CIN has complex causes including mitotic chromosome mis-segregation (*Thompson et al., 2010*), homologous recombination defects (*Li and Heyer, 2008*), telomere crisis (*Maciejowski et al., 2015*; *Maciejowski et al., 2020*), breakage-fusion-bridge cycles (*Gisselsson et al., 2000*), DNA replication stress (*Burrell et al., 2013*; *Bester et al., 2011*; *Tamura et al., 2020*), as well as others. Improving outcomes in HGSOC will depend on having well-characterized and validated pre-clinical *in vitro* models that accurately represent the CIN patterns observed in patients. However, currently available 2D models have multiple shortcomings such as changes in cell morphology, loss of diverse genotype and polarity, as well as other limitations. Patient-derived organoids (PDOs) offer improved pre-clinical cancer models and generally are molecularly representative of the donor, have good clinical annotation, and can represent tumoral intra-heterogeneity (*Vlachogiannis et al., 2018*; *Li et al., 2018*; *Lee et al., 2018*; *Kopper et al., 2019*). PDOs can be cultured for short periods (*Nelson et al., 2020*; *Hill et al., 2018*) but continuous HGSOC PDOs have only been generated for 27 models (*Kopper et al., 2019*; *Hoffmann et al., 2020*; *Maenhoudt et al., 2020*) and these models lack detailed genomic characterization to determine whether they adequately represent the genomic landscape of HGSOC.

Current pre-clinical models may significantly underrepresent common mutational processes observed in patients. Approximately 50% of HGSOC patients have impaired homologous recombination (HR) DNA repair, including approximately 15% of cases that have a loss of function and epigenetic events in *BRCA1* and *BRCA2* (*Cancer Genome Atlas Research Network, 2011*). Consequently, homologous-recombination deficiency (HRD) is the major genomic classifier in the clinic and stratifies patients for outcome after treatment with PARP inhibitors (*Gelmon et al., 2011*; *Swisher et al., 2017*). Despite the relatively high prevalence of HRD and *BRCA1/2* mutations in the clinic, there are only very few relevant models. This suggests that cell lines and PDOs that carry *BRCA1* and *BRCA2* deleterious mutations are selected against. In addition, there is an urgent unmet clinical need for therapies for patients with HGSOC that are homologous recombination proficient (HRP). Several distinctive patterns of structural variation have been described in HRP tumors including chromothripsis, tandem duplication (TD), whole-genome duplication (WGD), and *CCNE1* amplification (*Aaltonen et al., 2020*). Apart from the description of *CCNE1* amplification, it is unknown if the HGSOC organoids described to date display any of these genomic features and most cell line publications only refer to *BRCA1* and *BRCA2* mutations. These shortcomings highlight the lack of a systematic approach to characterize CIN and copy number signatures in PDO models.

To address these challenges, we developed HGSOC PDOs and characterized their genomes, transcriptomes, drug sensitivity, and intra-tumoral heterogeneity. Using copy number signatures, we show that our models comprehensively recapitulate clinically relevant genomic features across the whole spectrum of CIN observed in HGSOC patients. PDOs showed strong copy number-driven gene expression and transcriptional heterogeneity. Drug sensitivity was reproducible compared to parental tissues and the ability of these models to grow *in vivo*. Single-cell DNA sequencing showed copy number features at a subclonal level and distinct clonal populations. The PDO models we present

thus shed light on the ongoing chromosomal instability of HGSOC and can have clinical relevance for guiding treatment decisions.

## Results

### HGSOC organoid culture derivation

To establish HGSOC organoids we used cells obtained from patient-derived ascites (n=43), solid tumors (n=10), and patient-derived xenografts (n=15) (*Figure 1a*). Most ascites cultures were derived from patients with recurrent HGSOC and clinical summaries are provided in *Figure 1—figure supplement 2* and *Supplementary file 1*. We tested the effect of two published (*Kopper et al., 2019*; *Kessler et al., 2015*) media compositions on 15 independent cultures and found similar PDO viability (*Figure 1—figure supplement 1a*). We, therefore, performed subsequent derivations using the less complex fallopian tube media (*Kessler et al., 2015*). The efficiency of establishing PDOs was dependent on the type of tissue sample used for derivation (p<0.0001, log-rank test; n=86; *Figure 1—figure supplement 1b*) and the highest success rate for short-term cultures (passage number between 1 and 4) was obtained using ascites and dissociated xenograft tissues (65%). We defined continuous PDO cultures as those that could be serially passaged >5 times followed by cryopreservation and successful re-culture; all data in this paper was generated between passages 5–15. Using these criteria, PDOs were established for 15/18 organoid lines (PDO16, PDO17, and PDO18 were finite culture models). Four PDOs were able to grow as continuous 2D cell lines in conventional tissue culture media (CIOV7 from PDO1; CIOV5 from PDO2; CIOV4 from PDO3; and CIOV6 from PDO7).

PDOs were screened for mutations enriched in HGSOC using an in-house tagged amplicon sequencing panel (*Figure 1—figure supplement 3* and *Supplementary file 2*) and were highly comparable to mutational profiles and p53 immunostaining from the original patient sample (*Figure 1—figure supplement 4*). All PDOs had a *TP53* mutation allele fraction between 80–95% essentially excluding co-culture of non-cancer cells. Pathogenic somatic *BRCA1* or *BRCA2* mutations were present in PDO4, PDO7, PDO8, and PDO9. Germline DNA sequencing for 11 of the PDO donors (*Supplementary file 3*) showed *BRCA1/2* germline mutations with unknown clinical significance or benign variants in patients OV04-297 (PDO13), OV04-409 (PDO14), and OV04-627 (PDO5 and PDO6).

To assess the feasibility of the PDOs for *in vivo* modeling, we implanted eight PDO models into immunodeficient mice using intraperitoneal injection to simulate peritoneal metastasis. All eight PDOs efficiently established PDX models and 7/8 resulted in solid implants on peritoneal surfaces and/or liver infiltration (*Figure 1—figure supplement 5*).

### Genomic characterization of patient-derived organoids

We characterized the genomic landscape of the PDOs using sWGS and derived copy number signatures to characterize the diversity of causes of CIN (*Figure 1b* and *Figure 1—figure supplement 6*). We used our published framework for copy number signature extraction (*Macintyre et al., 2018*) based on non-negative matrix factorization (NMF) of feature-summarized copy number data to find the mutational processes behind the observed copy number profiles. We used the seven previously identified copy number signatures in ovarian cancer that represent different putative causes of CIN: s1: mitotic errors, s2: replication stress causing tandem duplication, s3 and s7: homologous recombination deficiency, s4: whole-genome duplication, s5: unknown etiology leading to chromothripsis, and s6: replication stress leading to focal amplification. The finite lines PDO16, PDO17, and PDO18 are included here for comparison only.

PDO1 and PDO11 showed high levels of signature s1 and are thus appropriate models of mitotic errors. PDO4 exhibited high activity of a signature of replication stress-induced tandem duplication (s2) but did not have a canonical *CDK12* mutation suggesting this may represent an alternative model of tandem duplication (see also below) (*Menghi et al., 2016*; *Willis et al., 2017*). Thirteen of the organoids showed evidence of s3 and can be considered as having HRD. Of these, pathogenic somatic *BRCA1* and *BRCA2* mutations were present in PDO4, PDO7, PDO8, and PDO9 (*Figure 1—figure supplement 3* and *Supplementary file 2*); a novel non-synonymous secondary mutation was observed in *BRCA1* (c.1367T>C) in PDO8 which was cultured after progression on PARP inhibitor therapy (paired with PDO7); *BRCA1/2* mutations were not detected in the remaining PDOs with s3 (PDO2, PDO3, PDO10, PDO12, PDO15) suggesting these may be models of other mechanisms of

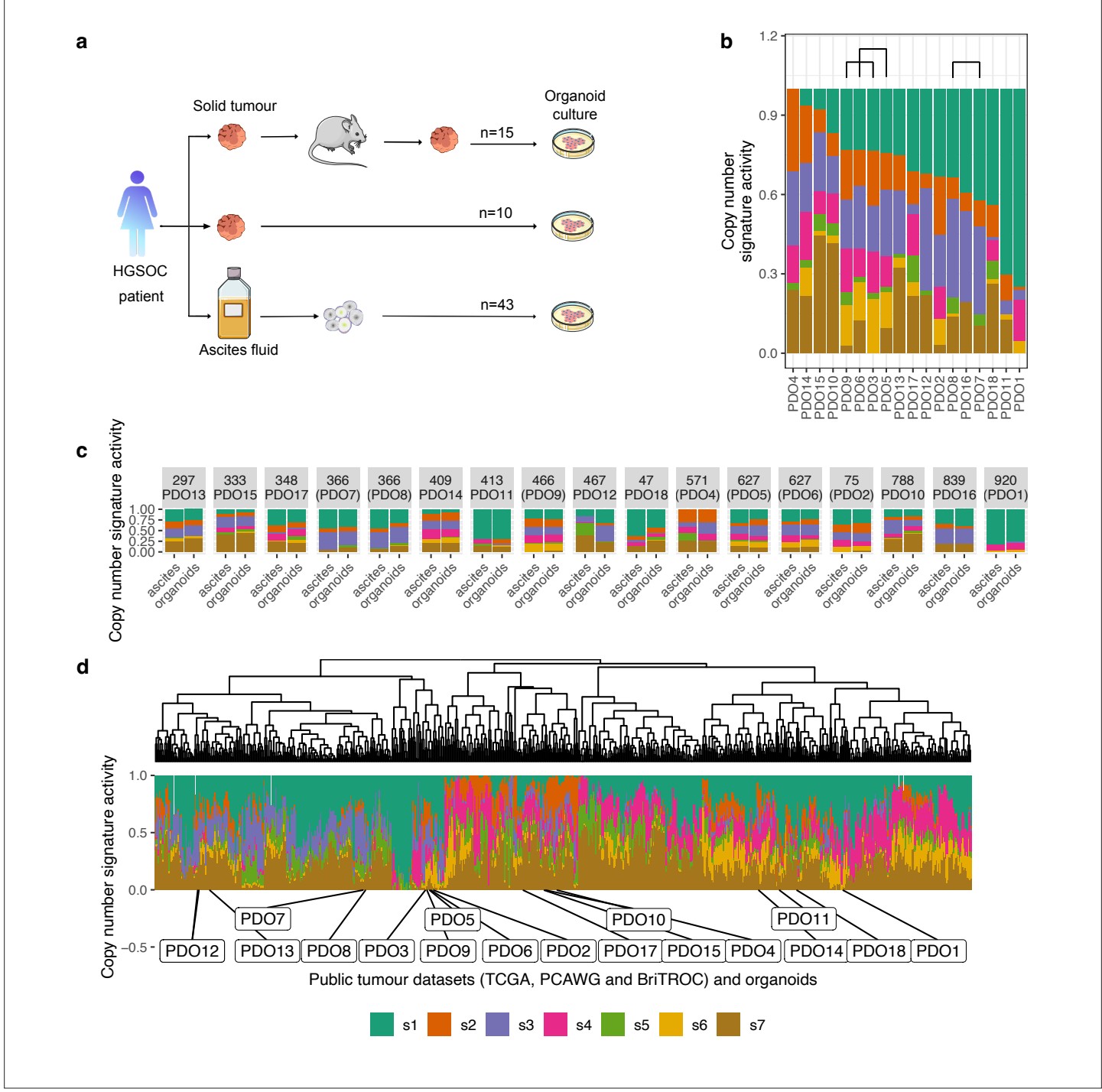

**Figure 1.** Chromosomal instability features of patient-derived organoids (PDOs). (**a**) Schematic of the sample collection workflow used in this study. (**b**) Stacked bar plots show copy number signature activities ranked by signature s1 (PDO16, PDO17, and PDO18 were not continuous models). Brackets indicate PDOs derived from the same individual. (**c**) Stacked bar plots show copy number signature activities for organoids and the matched ascites sample from which they were derived. (**d**) Unsupervised hierarchical clustering of copy number signature for PDO and 692 high-grade serous ovarian carcinoma (HGSOC) cases using Aitchison's distance with complete linkage function. Stacked barplots in the lower panel show copy number signature activities.

The online version of this article includes the following figure supplement(s) for figure 1:

**Figure supplement 1.** Organoid survival analysis.

**Figure supplement 2.** Clinical data.

*Figure 1 continued on next page*

HRD. PDO1 and PDO11 showed low signature s3 activity making them suitable models for HRP ovarian cancer. Ten of the PDOs showed s4 activity making them suitable to study the effects of WGD. Signature s5, with unknown etiology that results in chromothripsis, had generally low activity in all PDOs consistent with previous observations suggesting that canonical chromothripsis is a rare event in HGSOC (*Cortés-Ciriano et al., 2020*; *Zack et al., 2013*; *Patch et al., 2015*). s6, a signature of replication stress resulting in focal amplification, was high in PDO3, PDO5, PDO6, PDO9, and PDO14, indicating these are good models to study both the cause and consequence of focal amplification events. Finally, a number of organoids showed s7 making them good models to study the effects of HRD following WGD.

## Organoids represent the spectrum of human high-grade serous ovarian cancers

We next compared copy number signatures from donor patient tissues and matched PDO (*Figure 1c*) and found that they were highly consistent except for PDO12 (OV04-467). We tested for the differential abundance of the signatures between donor samples and matching PDO using previous described statistical modeling (*Cheng et al., 2022*). For the patients who contributed two samples, a single sample was selected at random. The results indicated no differential abundance (Wald test on log-ratios of signatures, p-value=0.99 using a model with no correlations between signatures given that the total number of observations is low). For PDO12, the parental *CDK12* mutation present in the ascites specimen was not recovered after culture, suggesting selection for a subclonal population with distinct copy number signatures (*Supplementary file 2*).

Both PDO culture and derivation of PDX models may negatively select against specific molecular subtypes of HGSOC—which may explain the low number of *BRCA1/2* models. To test whether the PDOs were representative of the wider population of HGSOC cases, we compared PDO copy number features to those of publicly available patient cohorts (n=692 samples from the TCGA, PCAWG, and BriTROC-1 studies) (*Figure 1d*). The number of copy number segments (*Figure 1—figure supplement 7a*) did not significantly differ between PDOs (169 ± 77) and HGSOC tissues from TCGA, PCAWG, and BriTROC-1 (200 ± 134) (p=0.22, negative binomial likelihood ratio test). Ploidy was found to be bimodal in both groups, with centers at average ploidies 2 and 3.5 (*Figure 1—figure supplement 7b*). There were also no significant differences in other copy number features (*Figure 1—figure supplement 7c*).

We next clustered copy number activity profiles (*Figure 1d*) from TCGA, PCAWG, and BriTROC (n=692) and compared these with the PDO profiles. Unsupervised hierarchical clustering of the clinical samples showed two main groups with the major group characterized by high activities for s4 and low activities for s3 suggesting frequent WGD and consistent with previous observations (*Aaltonen et al., 2020*; *Cheng et al., 2022*). The smaller group was predominantly composed of s1 mitotic errors and s3 HRD and may represent near diploid tumors. PDOs were well distributed across the two groups but there were three small subclusters that were underrepresented: those presenting a lack of s2 and s4, a lack of s2 and s3, and a lack of s3 together with high s4. PDOs derived from the same patient (PDO3 and PDO9, PDO5 and PDO6, and PDO7 and PDO8) were clustered together. Taken together, these data indicate that PDOs represents the copy number mutational landscape observed in HGSOC patients.

## Effect of CNAs at the gene expression level

To understand how PDO absolute copy number alterations (CNAs) could alter the gene expression of corresponding genes, we first tested whether PDOs displayed known HGSOC-associated amplifications (*Figure 2a*) and which genes were highly amplified when averaged over all PDOs

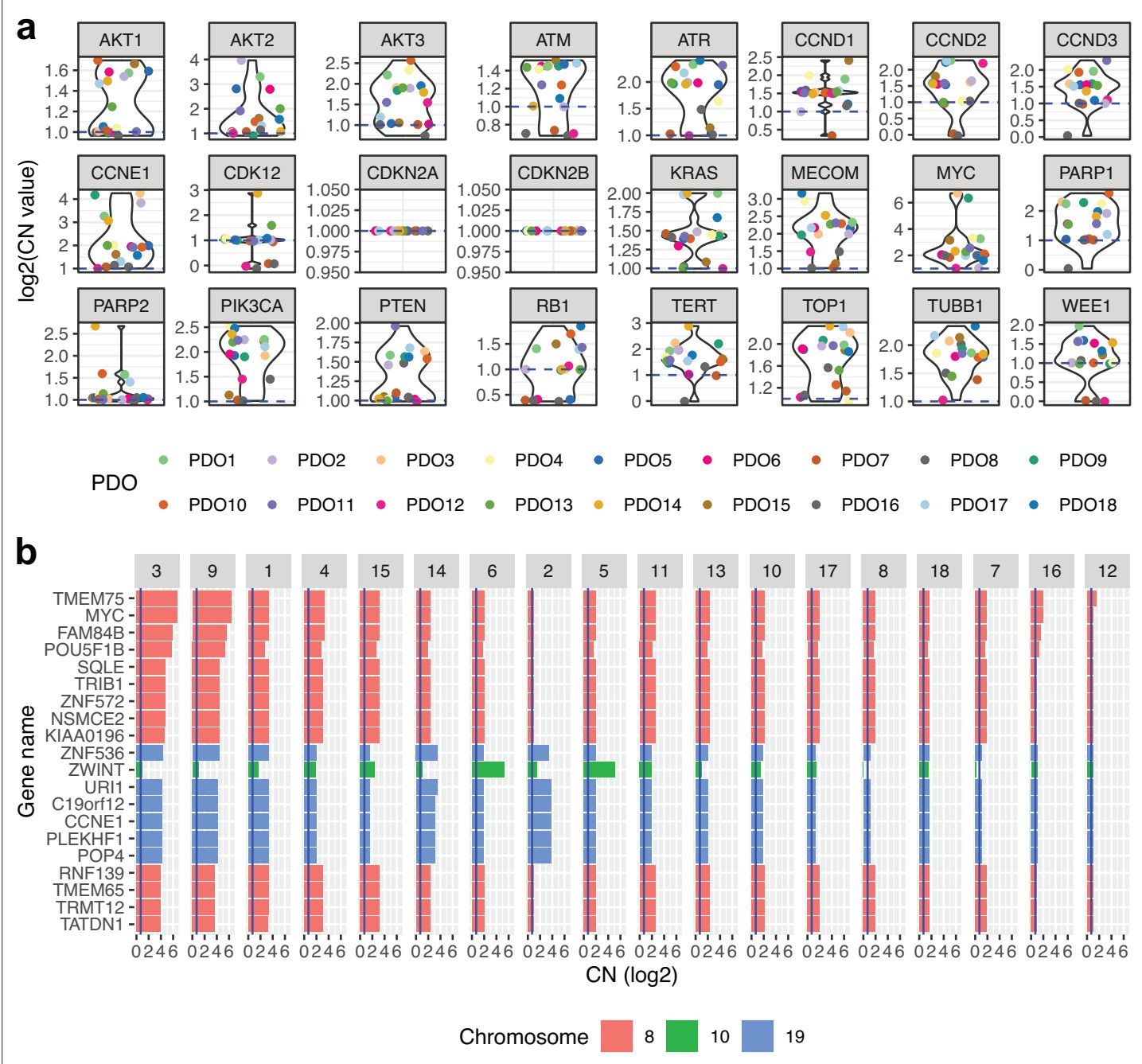

**Figure 2.** Absolute gene copy number in patient-derived organoids (PDOs). (**a**) Absolute gene copy number for a set of important high-grade serous ovarian cancer genes. (**b**) Absolute gene copy number for the most amplified genes when averaged across all patient-derived organoids.

The online version of this article includes the following figure supplement(s) for figure 2:

**Figure supplement 1.** Whole genome correlation between absolute gene copy number and expression.

(*Figure 2b*), including the well-characterized copy number drivers *MYC* and *CCNE1*. We performed RNA-Seq on the PDOs and compared their transcriptome to the TCGA primary tissue cohort and found highly similar cell-autonomous transcriptional profiles. As expected, we observed significant under-expression of genes relating to the tumor microenvironment (*Figure 3a*) which is not represented in the organoid cultures. Principal component analysis on the scaled and centered DESeq2 counts showed that PDOs derived from the same patient PDO5 and PDO6 - the transcriptome of which is nearly identical - cluster together, but that PDO7 and PDO8, which are distinguished

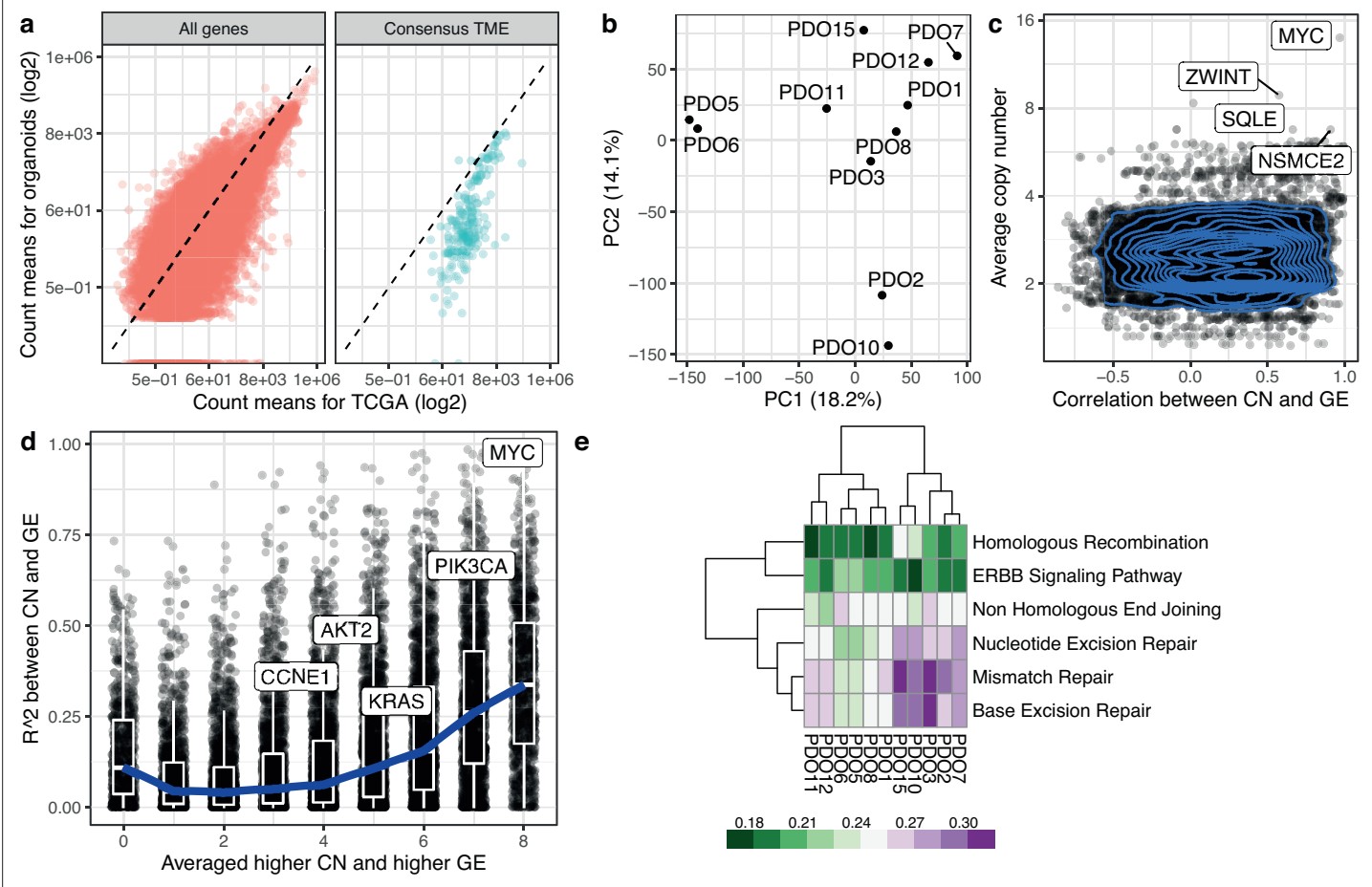

**Figure 3.** Transcriptomic analysis of high-grade serous ovarian carcinoma (HGSOC) organoids. (**a**) Scatterplots show correlation for the average counts, in transcripts per million (TPM) for each gene in the TCGA and the patient-derived organoid cohorts. Consensus TME genes represent non-tumor genes expressed in the tumor microenvironment (*Jiménez-Sánchez et al., 2019*). The dashed line corresponds to the identity line. (**b**) Principal component analysis based on DESeq2 counts for 11 organoids. (**c**) Scatterplot and contour plot of the Pearson correlation coefficient for copy number and gene expression, and average absolute copy number for each gene. *MYC* and *ZWINT* are shown as highly correlated genes. (**d**) Scatterplot of two metrics for assessing the agreement between copy number and gene expression. For each gene, we computed the average expression of the three organoids with the lowest copy number value. The metric is the fraction of remaining organoids that have higher gene expression value than this average, and takes values between 0/8 and 8/8, with higher values indicating greater agreement between copy number and gene expression across organoids. This is shown in the x-axis. On the y-axis we display the $R^2$ value for the correlation between copy number state and gene expression. We have labeled genes of interest. The blue curve indicates the median $R^2$ values in each group of the metric along the x-axis, and boxplots indicate the interquartile range. (**e**) DNA damage response KEGG pathway analysis from RNA-Seq on 11 PDOs. PDO10, and PDO15 show high enrichment scores for homologous recombination compared to other PDOs. Mismatch and base excision repair pathways also show high scores in these models. PDO8, which has the lowest HR score, contains a loss of function mutation in *BRCA1*.

by a secondary *BRCA1* mutation following progression after PARP therapy, differ from each other (*Figure 3b*). As PDOs are characterized by high *TP53* allele fractions in line with those seen in patient tumors, strongly indicating that they mostly consist of tumor cells, we assessed the correlation between gene copy number changes and their expression using two metrics. The first metric shows whether, on average, PDOs with lower copy number values in genes have a lower gene expression, in order to capture nonlinear relationships between copy number and gene expression. We computed the average gene expression values for the three PDOs of the lowest copy number and calculated the fraction of remaining PDOs with higher gene expression values than this average (*Figure 3c*). The second metric used was the $R^2$ of the correlation between DESeq2 count values and absolute copy number in each gene across PDOs. For both metrics, higher values indicate stronger evidence for copy number-driven gene expression (*Figure 3d*). The most highly variable areas in the genome are located within chromosomes 8, 10, 11, 12, 17, and 1 (*Figure 2—figure supplement 1a*), where we

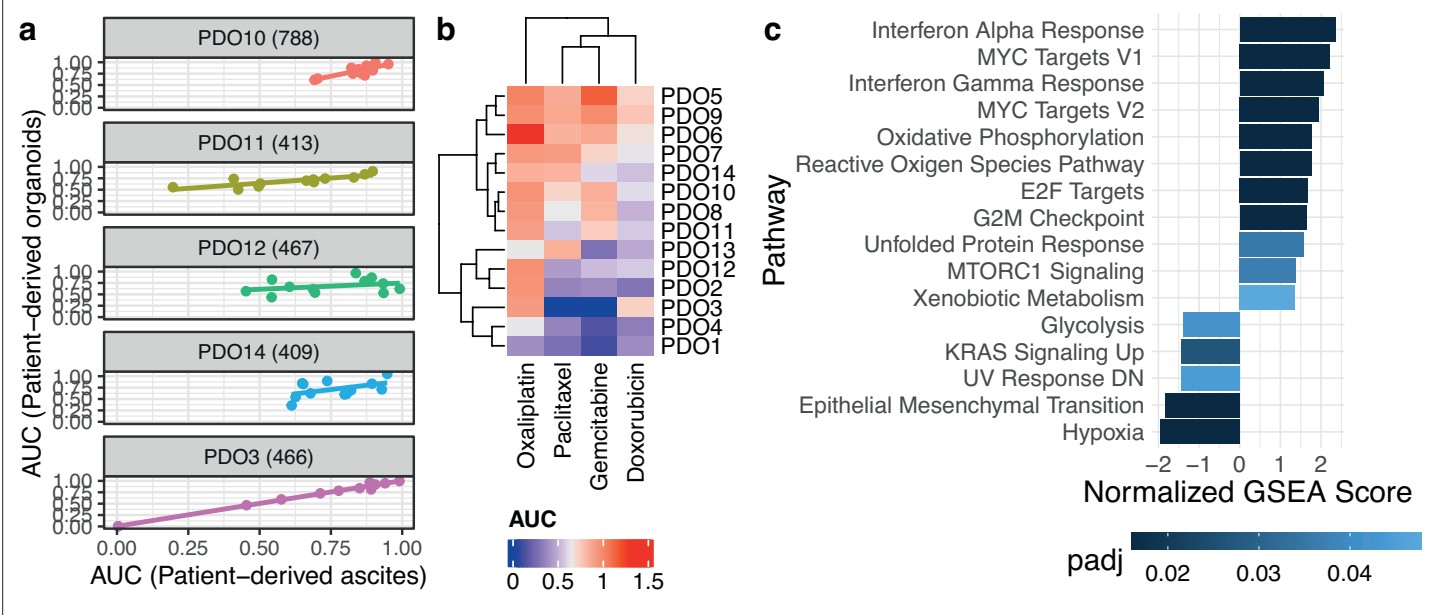

**Figure 4.** Patient-derived organoids are clinically relevant models. (**a**) Correlation of drug response between uncultured patient cells and the patient-derived organoids (PDOs) derived from them using 12 compounds (PDO14: cor. 0.49, p-value 0.1; PDO11: cor. 0.82, p-value 0.001; PDO3: cor. 0.995, p-value 2.3e-11; PDO10: cor. 0.81, p-value 0.001; PDO12: cor.0.32, p-value 0.31). (**b**) Organoid drug responses to standard-of-care chemotherapies. The observed dose-response relationships were not always compatible with the Hill dose-response model assuming a sigmoidal decrease so that five-parameter logistic model fits were preferred, explaining area under the curve (AUC) estimates greater than one. Sensitive PDOs are labeled with a blue dot and resistant PDOs with a red one. (**c**) Significant pathways based on adjusted p-value (padj) after performing Gene Set Enrichment Analysis (GSEA) with rank based on significance level between the two PDO groups sensitive and resistant.

The online version of this article includes the following figure supplement(s) for figure 4:

**Figure supplement 1.** Patient-derived organoids (PDOs) can be classified into two groups according to their drug sensitivity.

found the most highly correlated genes. *MYC* showed a good correlation between copy number and gene expression and was also the gene with the highest absolute copy number in our PDO cohort, followed by *ZWINT* (*Figure 2b*).

As defects in DNA damage response pathways are clinically important for treatment, we tested for enrichment scores across the PDOs. PDO10 and PDO15 have a high enrichment score for homologous recombination deficiency (*Figure 3e*), present nearly identical signature activities, and are the two PDOs with the highest s7 activity (*Figure 1b*).

## PDO drug screening

We compared drug sensitivity between five PDOs and their parental uncultured patient-ascites. Using 12 anti-cancer compounds dispensed in an 8-point half-log dilution series, we found a moderate to a high correlation between the drug area under the curve (AUC) of PDO and their corresponding patient-derived ascites (*Figure 4a*). We then tested all the PDOs using the standard of care chemotherapy (oxaliplatin, paclitaxel, gemcitabine, and doxorubicin) (*Figure 4b*) as we observed no effect with the targeted therapies at the concentrations used in this study. Based on the median AUC we divided PDOs into two groups of samples passing RNA-Seq quality control: sensitive (PDO1, PDO2, PDO3, PDO11, PDO12) and resistant (PDO5, PDO6, PDO7, PDO8, PDO10) (*Figure 4—figure supplement 1*) and performed differential gene expression and pathway analysis (*Figure 4c*) to infer mechanisms of resistance. Sensitive PDOs showed increases in *MYC* targets and interferon alpha and gamma responses while resistant PDOs had an increase in hypoxia, *KRAS* signaling, and epithelial-mesenchymal transition (EMT) pathways. We compared both groups for ploidy and number of copy number segments and we did not observe any significant differences (the average number of segments is 167 for sensitive and 183 resistant PDOs, and the average ploidies are 2.8 for sensitive and 2.46 for resistant PDOs; p-value=0.7441 and p-value=0.2374, respectively; Welch Two Sample t-test).

## Organoid intratumoral heterogeneity

In order to assess genomic heterogeneity within PDOs, we performed single-cell whole genome sequencing on three of the models, selected arbitrarily to represent both fast-growing (PDO2, n=76 cells, and PDO3, n=145 cells) and slow-growing models (PDO6, n=355 cells) (*Figure 5*). We did not observe any normal copy number profiles indicating the presence of non-cancer cells. Copy number changes at single-cell resolution revealed widespread clonal loss of heterozygosity (LOH) in large regions spanning up to entire chromosomes that were PDO specific (e.g. chromosome 13 in PDO6). Subclonal LOH, although less common, was also present in all three organoids. Amplification events were more common than losses; for example, chromosomes 2, 3, and 20 are clonally amplified in PDO2 and PDO3 whereas chromosomes 6 and 11 showed large, amplified regions shared between PDO3 and PDO6. All three PDOs present non-focal amplifications in chromosomes 1, 5, 12, and 20 as well as deletions in chromosome 13. This analysis also provided strong evidence for clonal amplification of candidate driver copy number aberrations: *CCNE1* in PDO2 and PDO3, an early chromothriptic event at *MYC* in PDO3 (*Figure 5—figure supplement 1*), and *AKT2* in PDO2 and PDO6. PDO6 showed early clonal loss of *RB1*.

We also identified regions of clonal heterogeneity in all three PDOs (*Figure 5* and *Figure 5—figure supplement 2*). We quantified the heterogeneity observed in each PDO by comparing the observed copy number variance to the expected copy number variance (Methods), and found that, globally, PDO3 showed the highest subclonal heterogeneity, with 48% of the genome presenting subclonal heterogeneity, followed by PDO6 (29%) and PDO2 (26%) (*Figure 5—figure supplement 3*).

## Discussion

Our analysis of copy number features and mutational signatures shows that HGSOC PDOs recapitulate the broad mutational landscape of patient samples. The organoid models contained a mixture of signatures indicating the influence of multiple mutational processes. Although their copy number signatures are well spread across the range seen in patient samples, certain copy number combinations are underrepresented (high s4 and s7, high s3 and s5, and high s6). Critically, we show that PDOs are also vital models to study heterogeneity at the single-cell level and we found that, although all models tested showed genomic heterogeneity, the level of complexity varies. This suggests that different mutational processes may have different abilities to drive evolutionary change and PDOs now provide tools for lineage tracing experiments to test this. Further analysis of clonal populations with PDO also has the potential to define the active mutational processes by sequential single-cell cloning as recently described (*Petljak et al., 2019*). Lastly, these models also provide important insights into the genomic etiology of HGSOC, including evidence for chromothripsis as an early initiation event in HGSOC by targeting *MYC* and indicating that tandem duplication can occur in the absence of either *BRCA1* or *CDK12* mutation.

The development of high-quality pre-clinical tumor models is of high importance for therapeutic discovery in HGSOC. Existing cell-based and PDX models have not been characterized in detail and their relationship to the diversity of CIN seen in patient tissue samples is unknown. Derivation of continuous cell lines has proven difficult for HGSOC, and although new cell lines are being developed (*Thu et al., 2017*; *Létourneau et al., 2012*; *Fleury et al., 2015*) success rates are comparatively low and the number of available models has not significantly increased over the past 10 years. With the wider use of organoid culture, ovarian cancer models have been developed both as short and long-term cultures (*Kopper et al., 2019*; *Nelson et al., 2020*; *Hill et al., 2018*; *Hoffmann et al., 2020*; *Maenhoudt et al., 2020*) but with variable information about success rates and survival in culture. We demonstrated that short-term HGSOC organoid derivation from human ascites samples and PDX tissues can be achieved with good efficiency. However, as indicated by our time-to-event analyses, further improvements in media and culture conditions are needed to improve success rates, particularly from solid tissue samples.

Although SCNAs have been shown to affect gene expression levels for the most abundantly expressed human genes indicating global gene dosage sensitivity (*Fehrmann et al., 2015*), it has also been described that this correlation does not always translate proportionally due to transcriptional adaptive mechanisms (*Bhattacharya et al., 2020*). In our study we compared PDO gene expression to TCGA patient samples and corroborated that gene transcript levels are highly correlated, providing ideal models to study tumor cell-intrinsic associations. We have previously found that the correlation between SCNA and gene expression is higher for cancer driver genes that are frequently amplified and

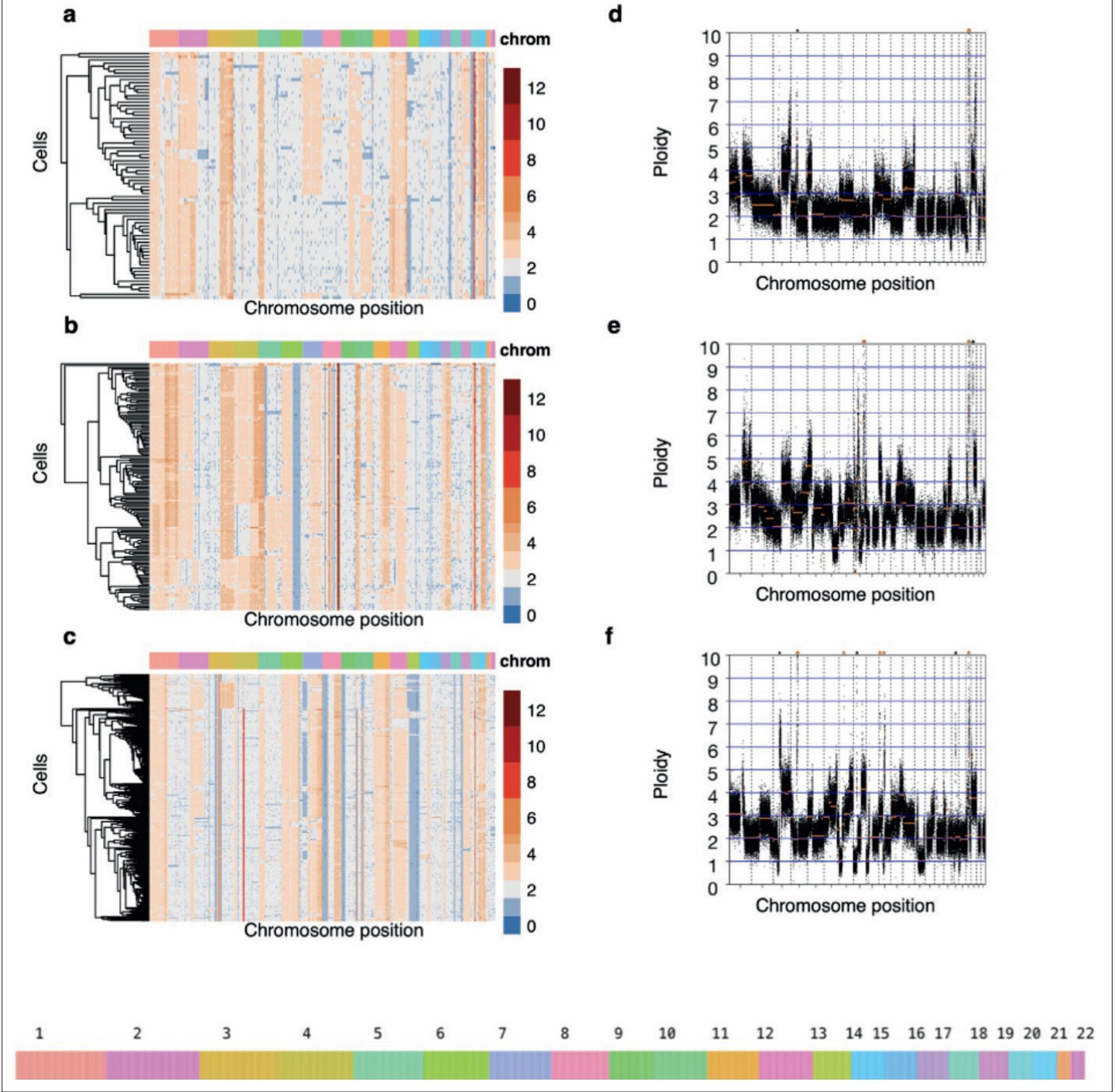

**Figure 5.** Genomic heterogeneity in three high-grade serous carcinoma patient-derived organoids (PDOs). (**a–c**) Single-cell DNA (scDNA) copy number where cells have been clustered using hierarchical clustering on Euclidean distance. Each row within the scDNA plots represents a cell across the different chromosomes in the x-axis and the copy number state (20 kb bins) is indicated in colors. Loss of heterozygosity and amplification events are common in all three patient-derived organoids. (**d-f**) Bulk absolute copy number profiles.

The online version of this article includes the following figure supplement(s) for figure 5:

**Figure supplement 1.** Chromothripsis in chromosome 8 of PDO3.

**Figure supplement 2.** Major clades of cells in three organoids determined by copy number alterations from single-cell DNA-Seq.

**Figure supplement 3.** 95% confidence intervals of the centered copy number, along the genome, across single-cells from organoids.

identified co-dependencies between amplification of *MYC* and genes from the *PI3K* pathway which have therapeutic potential (*Martins et al., 2022*). We corroborated, using novel ways of correlating absolute SCNA with transcriptomics, that in our organoid models, the correlation was highest for *MYC*, *PIK3CA*, and *AKT2* reinforcing their putative role as potential targetable cancer drivers.

Genetic alterations in HGSOC are extraordinarily diverse therefore the development of a truly personalized treatment requires genomically annotated individual patient avatars for therapeutics. In this study, we showed the potential of HGSOC PDOs as a new preclinical cancer model representing individual patients. Consistent with studies in ovarian cancer and other tissue types (*Lee et al., 2018*; *Gao et al., 2014*; *Broutier et al., 2017*; *Francies et al., 2016*) our results confirm the feasibility of using PDOs for testing drug sensitivity in HGSOC. Future studies should account for doubling-time confounding errors using different metrics such as Growth Rate (GR) metrics (*Hafner et al., 2016*).

This study has shown that HGSOC PDOs faithfully represent the high variability in copy number genotypes observed in HGSOC patients and together with their associated clinical, phenotypic, and genomic characterizations will provide an important resource for pre-clinical and translational studies investigating genomic biomarkers for treatment stratification and further our understanding of tumor heterogeneity and clonality.

## Methods

### Ethical approval and clinical data collection

Clinical data and tissue samples for the patients were collected on the prospective cohort study Cambridge Translational Cancer Research Ovarian Study 04 (CTCR-OV04), with IRAS project ID 4853, and which was approved by the Institutional Ethics Committee (REC reference number 08 /H0306/61). Clinical decisions were made by a clinical multidisciplinary team (MDT) and researchers were not directly involved. Patients provided written, informed consent for participation in this study and for the use of their donated tissue for the laboratory studies carried out in this work and its publication. Clinical data for all the patients is provided in Supplementary Information.

### Sample collection and processing

Samples were obtained from surgical resection, therapeutic drainage, or surgical washings. Solid tumors were assessed by a pathologist and only tumor samples with ≥50% cellularity were attempted to grow. A small portion of each sample was kept at −80 °C until used for genomic profiling.

### Organoid derivation

Tumor samples were washed in PBS, minced into 2 mm pieces using scalpels, and incubated with gentamicin (50 µg/ml), Bovine Serum Albumin Fraction V (1.5%), insulin (5 µg/mL), collagenase A (1 mg/mL) and hyaluronidase (100 U/ml) for 1–2 hr at 37 °C. Following incubation, the mixture was filtered and the cell suspension was spun down and washed with PBS. Ascites fluid was centrifuged at 450 g for 5 min. Cells were then washed with PBS and centrifuged at 400 g for 5 min.

The isolated cells were resuspended in 7.5 mg/ml basement membrane matrix (Cultrex BME RGF type 2 (BME-2), Amsbio) supplemented with complete media and plated as 20 µl droplets in a six-well plate. After allowing the BME-2 to polymerize, complete media was added and the cells were left at 37 °C. We used published culture conditions for normal fallopian tube growth (*Kessler et al., 2015*) as follows: AdDMEM/F12 medium supplemented with HEPES (1×, Invitrogen), Glutamax (1×, Invitrogen), penicillin/ streptomycin (1×, Invitrogen), B27 (1×, Invitrogen), N2 (1×, Invitrogen), Wnt3a-conditioned medium (25% v/v), RSPO1-conditioned medium (25% v/v), recombinant Noggin protein (100 ng/ml, Peprotech), epidermal growth factor (EGF, 10 ng/ml, Peprotech), fibroblast growth factor 10 (FGF10, 100 ng/ml, Peprotech), nicotinamide (1 mM, Sigma), SB431542 (0.5 µM, Cambridge Biosciences), and Y27632 (9 µM, Abmole).

### Organoid culture

Organoid culture medium was refreshed every 2 days. To passage the organoids, the domes were scraped and collected in a falcon tube, TrypLE (Invitrogen) was added and incubated at 37 °C for approximately 10 min. The suspension was centrifuged at 800 g for 2 min and the cell pellet was resuspended in 7.5 mg/ml BME-2 supplemented with complete media and plated as 20 µl droplets in a six-well plate. After allowing the BME-2 to polymerize, complete media was added, and cells

were incubated at 37 °C. The commonest cause of culture failure was growth arrest or fibroblast overgrowth. We considered an organoid line to be continuously established when it had been serially passaged >5 times followed by cryopreservation and successful re-culture. By these criteria, 15/18 PDO lines were continuous.

## Immunohistochemistry

Haematoxylin and Eosin (H&E) slides were stained according to the Harris H&E staining protocol and using a Leica ST5020 multi-stainer instrument. Paraffin-embedded sections of 3 µm were stained using Leica Bond Max fully automated IHC system. Briefly, slides were retrieved using sodium citrate for 30 min and p53 antibody (D07, 1:1000, Dako) was applied for 30 min. Bond Polymer Refine Detection System (Leica Microsystems) was used to visualize the brown precipitate from the chromogenic substrate, 3,3'-Diaminobenzidine tetrahydrochloride (DAB).

## Nucleic acid isolation

DNA and RNA were extracted at the same time from the same cells. Extraction was performed using the DNeasy Blood & Tissue Kit (QIAGEN) according to manufacturer instructions.

## Bulk shallow whole-genome sequencing and absolute copy number signature analysis

Whole genome libraries were prepared using the TruSeq Nano Kit according to manufacturer instructions. Each library was quantified using the KAPA Library Quantification kit (kappa Biosystems) and 10 nM of each library was combined in a pool of 21 samples and sequenced on the Illumina HiSeq 4000 machine using single-end 150 bp reads. Reads were aligned against the human genome assembly GRCh37 using the BWA-MEM algorithm (v0.7.12). Duplicates were marked using the Picard Tool (v1.47) and copy number was assessed using the Bioconductor package QDNAseq (v1.6.1) (*Scheinin et al., 2014*). Shallow whole-genome samples have an approximate coverage of 0.25–0.3, assuming that the sample is diploid.

Copy number signatures for the organoid cultures were calculated as previously described (*Macintyre et al., 2018*).

## Comparison of organoid copy number signatures to those of TCGA, BriTROC-1, and PCAWG

Signature activities of organoids were compared to those previously described in three HGSOC cohorts: TCGA and BriTROC-1 (*Macintyre et al., 2018*) (sWGS-based signatures) and PCAWG (*Aaltonen et al., 2020*) (WGS-based signatures). Copy number signature activities were transformed using the centered log-ratio transformation with an imputation value of $10^{-2}$ to consider that they are compositional data that sample-wise add up to one. Organoid and primary tissue samples were clustered using hierarchical clustering with complete linkage on this transformed space. We performed additional analyses to confirm that our conclusions – namely, that the signature activities of organoids are representative of the activities of primary tissue, and in determining which activities are underrepresented in the organoids – were robust to the imputation value. Using imputation values between 0.001 and 0.1 we show that the dendrogram in *Figure 1d* is similar to the dendrograms generated using both higher and lower imputation values, and that the underrepresented clades are robust to changes in the imputation values. A more detailed report of the differences in dendrograms as we vary the imputation values can be found in the GitHub repository (see below).

## Comparison of copy number signatures between ascites and organoids

Signature exposures between ascites and organoids are compared using the same method as in a previous CN paper (*Cheng et al., 2022*), in which the model is detailed. Briefly, the model used is a multivariate model on isometric log-ratio (ILR)-transformed exposures that accounts for data compositionality, by modeling these transformed quantities as a non-correlated multivariate normal distribution, and testing for a difference in the mean of the two groups.

## Single-cell sWGS

Organoids were dissociated into single cells using TrypLE, washed twice with PBS, and counted. Single-cell solution was filtered using a 70 µm Flowmi filter to remove any duplets or triplets. With

the aim of getting around 300 cells for library preparation, 4000 single cells were loaded onto the chip. Single-cell 10 x CNV libraries were prepared according to the manufacturer's protocol (10 X Genomics) and multiplexed in equal molarity to achieve 2.4 million reads per cell. Single-cell 10 X CNV constructed libraries were sequenced on the Illumina Novaseq6000 S4 platform using PE- 150 modes. The Cell Ranger pipeline was used for quality control, trimming, and alignment.

## Metric for copy number subclonal heterogeneity in single-cell

The metric for copy number subclonal heterogeneity is defined as follows. Independently, for each of the three organoids, we fitted a linear model of the standard deviation of the absolute copy number across organoids predicted by its mean, using bins of 500 kb. Copy number data were handled using the R package GenomicRanges (*Lawrence et al., 2013*). The marked positive correlation indicated that the data were heteroscedastic. For each bin, we computed its expected variance from the model, $E(\sigma^2)$, and compared it to the observed variance $S^2$ with a Chi-Squared test with alternative hypothesis $E(\sigma^2) < S^2$. A statistically significant result indicates that we see a greater variance than expected in the copy number values of this bin, and that, therefore, there is subclonal heterogeneity.

## Clade analysis of single-cell copy number data

Single-cell clades for each organoid were identified by performing hierarchical clustering using complete linkage on Euclidean distance of copy number values on 500 kb-binned genomes. Only clades with more than three cells were kept in the analysis. PDO2 had four major clades, two of which encompassed most cells (clade A: 42 cells, clade B: 30 cells), PDO3 had seven major clades, three of which with more than two cells (clade A: 40 cells, clade B: 52 cells, clade C: 48 cells). PDO6 had six clades, three of which contained more than one cell (clade A: 158 cells, clade B: 145 cells, clade C: 49 cells). The copy number profile comparison of the two clades of PDO2, and of the two pairwise comparisons of clades of PDO3 and PDO4, were carried out using the 20 kb-binned copy number profile. Bins of distinct copy numbers between cells in different clades were detected using a Holm–Bonferroni-adjusted *t*-test on the absolute copy number value.

## Tagged-amplicon sequencing

Coding sequences of *TP53, PTEN, NF1, BRCA1, BRCA2, MLH1, MSH2, MSH6, PMS2, RAD51C, RAD51B, RAD51D*, and hot spots for *EGFR, KRAS, BRAF, PIK3CA* were sequenced using tagged amplicon sequencing on the Fluidigm Access Array 48.48 platform as previously described (*Forshew et al., 2012*). Libraries were sequenced on the MiSeq platform using paired-end 125 bp reads. Variant calling from sequencing data was performed using an in-house analysis pipeline and IGV software (*Thorvaldsdóttir et al., 2013*).

## RNA-Seq

RNA quality control was performed using Tapestation according to manufacturer instructions and samples were processed using Illumina's TruSeq stranded mRNA kit with 12 PCR cycles according to manufacturer's instructions. Quality control of libraries was performed using Tapestation and Clariostar before normalizing and pooling. Samples were sequenced using two lanes of SE50 on a HiSeq 4000 instrument. The analysis was performed using an in-house DESeq2 (*Love et al., 2014*) pipeline.

TCGA gene expression values were downloaded as HTSeq count files of Genome Build GRCh38 for 240 ovarian samples of either progressive disease, or complete remission or response. The counts were normalized using the DESeq2 method, based on gene-specific geometric means. The subset of genes relating to the tumor microenvironment was taken from the Consensus^TME list (https://github.com/cansysbio/ConsensusTME, *Cast, 2023*). The normalized expression of all genes was used to create the PCA.

## Effect of CNAs at the gene expression level

We computed the average gene expression values for the three PDOs of the lowest copy number. Three organoids, out of eleven, with the lowest expression were selected in order not to include solely outliers, as well as leaving out a high enough number of organoid samples (eight) in which we can observe the variability in their copy number and gene expression. We explored using the two, and four, PDOs and the lowest copy number, which yielded similar results - there is a very high correlation

between these averaged GE values when using the lowest three organoids, and when using the lowest two, or four.

## Pathway enrichment analysis

Using our transcriptomic data, we computed enrichment scores for KEGG pathways of interest using ssGSEA, implemented in the R package GSVA (*Hänzelmann et al., 2013*), and using gene sets from the package GSVAdata (*Hänzelmann et al., 2013*). To determine which pathways were overrepresented in the differential expression analysis between sensitive and resistant samples we used the R package fgsea (*Korotkevich et al., 2021*) and selected the top ten pathways according to their adjusted p-value (Benjamini-Hochberg correction), using the Hallmark gene sets from MSigDBv5p2.

## Drug sensitivity

An eight-point half-log dilution series of each compound was dispensed into 384 well plates using an Echo 550 acoustic liquid handler instrument (Labcyte) and kept at –20 °C until used. Prior to use plates were spun down and 50 µl of organoid suspension is added per well using a Multidrop Combi Reagent Dispenser (Thermo-Fisher). Following 5 days of drug incubation cell viability was assayed using 30 µl of CellTiter-Glo (Promega). Screens were performed in technical triplicate.

Drug response measures were standardized by dividing the original values by the median drug response observed in the control group of each drug and sample and then modeled as a function of the dose (on the log scale) by means of a $4^{th}$-degree polynomial robust regression, fitted by means of the function lmrob of the R package robustbase (*Maechler et al., 2023*). Drug response measures that obtained robust weights smaller than 0.4 (out of a range which spreads from 0 for outliers to 1 for non-outliers) were considered as outliers. After excluding outliers, we modeled the standardized drug response measures as a function of the dose (on the log scale) by means of the five-parameter log-logistic model (drm function of the drc R package [*Ritz et al., 2015*] with fct argument set to LL2.5). Area under the curve estimates was finally obtained by integrating the expected standardized drug response given the dose on the dose range of interest (on the log scale). Note that the use of M-splines instead of a log-logistic model led to similar AUC estimates.

Compounds used in this study included standard-of-care chemotherapeutics paclitaxel (Sigma), oxaliplatin (Selleck), doxorubicin (Selleck), and gemcitabine (Selleck); and targeted compounds provided by AstraZeneca: AZD0156, AZD2014, AZD6738, AZD2281, AZD1775, AZD8835, AZD5363, and AZD8185. Maximum drug concentration in the assay was 30 µM apart from paclitaxel (0.3 µM) and oxaliplatin (300 µM).

## *In vivo* growth

Animal procedures were conducted in accordance with the ethical regulations and guidelines of AWERB, NACWO, and UK Home Office (Animals Scientific Procedures Act 1986). It was approved by the CRUK CI Animal Welfare and Ethics Review Board (Home Office Project Licence number: PP7478310). 1.5 × 105 organoids were resuspended in 150 µl of PBS and injected intraperitoneally into NOD-scid IL2Rγ(null) (NSG) mice. Tumor growth was monitored by palpation and weighing the mice weekly.

## Code availability

All the analysis code is at https://github.com/lm687/Organoids_Compositional_Analysis (copy archived at *Morrill, 2023*).

## Acknowledgements

We thank all patients who participated in and donated tissue samples to this study. The Addenbrooke's Human Research Tissue Bank is supported by the NIHR Cambridge Biomedical Research Centre. We also thank Karen Hosking, Mercedes Jimenez-Linan, and the OV04 study team for their help with clinical tissue samples. We also thank staff from the Cancer Molecular Diagnostics Laboratory for performing blood and ascites collections. We would like to thank the Cancer Research UK Cambridge Institute Genomics, IT & Scientific Computing, Biological Resource Unit, Compliance & Biobanking, Research Instrumentation and Cell Services, and Bioinformatics core facilities for their support with various aspects of this study. The results shown here are in part based upon data generated by the TCGA Research Network: https://www.cancer.gov/tcga. The views expressed are those of the authors

and not necessarily those of the NIHR or the Department of Health and Social Care. We thank Susana Ros, Thomas Bradley, and Hayley Frances for critically reading the manuscript. We would like to thank the Clevers Laboratory (University of Utrecht) for hosting Maria Vias for a CRUK Travel Award. We thank the reviewers and the editors.

## Additional information

### Competing interests

Anna M Piskorz, Florian Markowetz, Geoff Macintyre, James D Brenton: GM, FM, AMP and JDB are founders and shareholders of Tailor Bio Ltd. The other authors declare that no competing interests exist.

### Funding

| Funder | Grant reference number | Author |
| --- | --- | --- |
| Wellcome Trust | RG92770 | Lena Morrill Gavarró |
| Marie Sklodowska-Curie Actions | 766030-CONTRA-H2020-MSCA-ITN-2017 | Michael P Schneider |
| Cancer Research UK Cambridge Institute, University of Cambridge | A22905 | James D Brenton |
| Cancer Research UK Cambridge Institute, University of Cambridge | A29580 | James D Brenton |
| Cancer Research UK Cambridge Institute, University of Cambridge | A25117 | James D Brenton |
| NIHR Cambridge Biomedical Research Centre | BRC-1215-20014 | James D Brenton |

The funders had no role in study design, data collection and interpretation, or the decision to submit the work for publication. For the purpose of Open Access, the authors have applied a CC BY public copyright license to any Author Accepted Manuscript version arising from this submission.

### Author contributions

Maria Vias, Conceptualization, Resources, Data curation, Formal analysis, Supervision, Validation, Investigation, Methodology, Writing – original draft, Writing – review and editing; Lena Morrill Gavarró, Conceptualization, Data curation, Software, Formal analysis, Validation, Investigation, Visualization, Methodology, Writing – original draft, Writing – review and editing; Carolin M Sauer, Resources, Data curation, Formal analysis, Investigation, Methodology, Writing – review and editing; Deborah A Sanders, Anna M Piskorz, James Hall, Resources, Investigation, Methodology; Dominique-Laurent Couturier, Formal analysis, Methodology, Writing – review and editing; Stéphane Ballereau, Data curation, Formal analysis, Methodology, Writing – review and editing; Bárbara Hernando, Data curation, Formal analysis, Writing – review and editing; Michael P Schneider, Data curation, Methodology; Filipe Correia-Martins, Resources, Data curation, Investigation, Methodology, Writing – review and editing; Florian Markowetz, James D Brenton, Conceptualization, Supervision, Writing – review and editing; Geoff Macintyre, Conceptualization, Formal analysis, Supervision, Methodology, Writing – review and editing

### Author ORCIDs

Maria Vias ⓘ http://orcid.org/0000-0003-4955-0102
Lena Morrill Gavarró ⓘ http://orcid.org/0000-0002-2242-4010
Carolin M Sauer ⓘ http://orcid.org/0000-0003-2168-6630
Michael P Schneider ⓘ http://orcid.org/0000-0002-6331-2357
James Hall ⓘ http://orcid.org/0000-0002-8124-5434

Florian Markowetz (iD) http://orcid.org/0000-0002-2784-5308
James D Brenton (iD) http://orcid.org/0000-0002-5738-6683

## Ethics

Human subjects: Clinical data and tissue samples for the patients were collected on the prospective cohort study Cambridge Translational Cancer Research Ovarian Study 04 (CTCR-OV04), with IRAS project ID 4853, and which was approved by the Institutional Ethics Committee (REC reference number 08/H0306/61). Clinical decisions were made by a clinical multidisciplinary team (MDT) and researchers were not directly involved. Patients provided written, informed consent for participation in this study and for the use of their donated tissue for the laboratory studies carried out in this work and its publication.

Animal procedures were conducted in accordance with the ethical regulations and guidelines of AWERB, NACWO and UK Home Office (Animals Scientific Procedures Act 1986). It was approved the CRUK CI Animal Welfare and Ethics Review Board (Home Office Project Licence number: PP7478310).

## Decision letter and Author response

Decision letter https://doi.org/10.7554/eLife.83867.sa1
Author response https://doi.org/10.7554/eLife.83867.sa2

---

# Additional files

## Supplementary files

• Supplementary file 1. Summary of patient chemotherapy treatment.

• Supplementary file 2. Mutation analysis of patient and PDO samples.

• Supplementary file 3. Patient germline BRCA mutation status.

• MDAR checklist

## Data availability

RNA-Seq data are available at the Gene Expression Omnibus (GEO) under accession number GSE208216, and sWGS and scDNA data are available at the EGA European Genome-Phenome Archive (EGA) under accession number EGAS00001007189. These data are available for non-commercial academic use only.

The following datasets were generated:

| Author(s) | Year | Dataset title | Dataset URL | Database and Identifier |
|---|---|---|---|---|
| Vias M, Gavarró LM, Sauer CM, Sanders D, Piskorz AM, Couturier D, Ballereau S, Hernando B, Hall J, Correia-Martins F, Markowetz F, Macintyre G, Brenton JD | 2022 | High grade serous ovarian cancer organoids as models of chromosomal instability | https://www.ncbi.nlm.nih.gov/geo/query/acc.cgi?acc=GSE208216 | NCBI Gene Expression Omnibus, GSE208216 |
| Vias M, Gavarró LM, Sauer CM, Sanders D, Piskorz AM, Couturier D, Ballereau S, Hernando B, Hall J, Correia-Martins F, Markowetz F, Brenton JD | 2023 | HGSOC organoid sequencing study | https://ega-archive.org/studies/EGAS00001007189 | European Genome-Phenome Archive, EGAS00001007189 |

The following previously published dataset was used:

| Author(s) | Year | Dataset title | Dataset URL | Database and Identifier |
|---|---|---|---|---|
| Macintyre et al. | 2017 | Copy number signatures and mutational processes in ovarian carcinoma | https://ega-archive.org/studies/EGAS00001002557 | European Genome-Phenome Archive, EGAS00001002557 |

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
