## [Editor Report]

This fundamental work substantially advances our understanding of patient-derived organoids as a useful model to evaluate chromosome instability and identify novel therapeutic strategies to combat HGSOC. The study is comprehensive, and the evidence supporting the conclusions is compelling, which would further benefit the related research about the mechanisms of genomic instability in HGSOC.

---

## [Decision Letter]

**Decision letter after peer review:**

Thank you for submitting your article "High-grade serous ovarian carcinoma organoids as models of chromosomal instability" for consideration by *eLife*. Your article has been reviewed by 3 peer reviewers, and the evaluation has been overseen by a Reviewing Editor and Tony Ng as the Senior Editor. The following individuals involved in the review of your submission have agreed to reveal their identity: Kundan Sengupta (Reviewer #1); Mark Nachtigal (Reviewer #3).

The reviewers have discussed their reviews with one another, and revisions are necessary for the manuscript. For your guidance, the reviewers' comments are appended below. If you decide to revise your manuscript, please revise your work guided by the reviewers' suggestions, and provide a point-by-point response to the following suggestions and concerns.

*Reviewer #1 (Recommendations for the authors):*

1. The basis for categorizing CIN into S1 through S7 needs to be explained with better clarity.

2. Page#1: The point that "recurrent somatic substitutions are rare and involve <10 driver genes", if these mutations involve mutator genes, this is likely to involve many additional genes. For instance, how would mutations in MSH2, or p53 contribute to additional mutations?

3. Page#3: While it is interesting to note that mutations were found in BRCA1/2, were mutations also detected across passages of the organoids in p53 and BRCA1/2?

4. Page#3, Line 79-81: Remarkably, all 8 PDOs resulted in solid implants. Did these PDX also recapitulate mutations as found in the PDOs?

5. Page#4: From the spectrum of mutations highlighted, it seems obvious that the PDOs show conserved mutations, while PDOs also show considerable heterogeneity. Would this finding then contradict the premise that PDOs clonally evolve and may not necessarily recapitulate mutations inherent in the HSCOGs?

6. Page#7: Do most PDOs show a ploidy of 2 and 3.5? Does the ploidy increase correlate with an increase in the diverse subtypes of CIN?

7. From this data, it is unclear how significant these changes/alterations in CIN are.

8. Figure#1: What is the extent of heterogeneity across the PDOs? this is not evident from this data.

9. Figure#2: Transcriptomic analyses of the PDOs reveals critical aspects of how expression profiles of HSCOGs correlate with their transcriptome. Is there a mechanistic basis for the enrichment of the HR against the NHEJ pathway in certain PDOs?

10. Figure#3: The PDO drug screening is an interesting and revealing experiment. However, the explanation is rather confusing, since the demarcation between sensitive and resistant PDOs does not include all the PDOs under each category. Figure 3A: Could include representatives from sensitive and resistant categories with a label accordingly for each of the two categories. Figure 3C: Is this data derived from each PDO from the sensitive and resistant categories? or is this pooled data? this is not clear from the Results section. Although the GSEA enrichment analyses enrich for MYC targets, E2F and G2/M checkpoint targets.

11. Figure#4: This figure is not numbered. It would be useful to indicate and explain each panel. Label the left panels as A-C and insert the legend on the X and Y-axis respectively. Likewise, label the columns on the right and label as D-F.

*Reviewer #2 (Recommendations for the authors):*

1. The authors performed single-cell whole genome sequencing on three PDOs (PDO2, PDO3, PDO6). It is not clear why the authors chose these three PDOs for single-cell whole-genome sequencing. This method is good for organoid intratumoural heterogeneity study if the authors provide more samples of single-cell whole genome sequencing.

2. In line 147, the authors said PDOs consist of 100% tumour cells, are you sure?

3. In the PDO drug screening part, the authors performed gene expression and pathway analysis on sensitive and resistant groups. How about the mutations and CNAs?

*Reviewer #3 (Recommendations for the authors):*

This was very challenging to read due to lack of information (e.g., Figure 1d, Supplementary Figure 3b, and Supplementary Table 4 were not included with the manuscript), and limited or lacking justification for the inclusion of data. Some of the figures do not adequately transmit the intended information and the figure legends should be revised to provide more comprehensive information to the reader. Justification for some analytical approaches (e.g., selection of gene expression and CN comparison groups) would enhance the manuscript. Please see specific comments below.

Organoids are typically composed of numerous cell types to recapitulate an organ, whereas spheroids are typically constituted of a single cell type grown in a 3-dimensional format. A weakness of the manuscript is the lack of evidence demonstrating the composition of the PDOs and the lack of an operational definition of an organoid. Without further evidence, these 3D cultures should be referred to as tumourspheres or spheroids. Some of the most informative research on the development of multicellular models to recapitulate high-grade serous ovarian cancer are from the Balkwill laboratory (see Malacrida at al. Building in vitro 3D multicellular models of high-grade serous ovarian cancer. STAR Protocols 3(1): 101086. DOI: 10.1016/j.xpro.2021.101086). In general, the epithelial ovarian cancer field accepts the definition of organoids as those with multiple cell types (e.g., tumour cells, mesothelial cells, and lymphocytes).

CIN is a type of genomic instability that is broadly defined as an increased rate of chromosome gains or losses and is best identified through analysis of single cells (e.g., karyotype analysis), something that bulk whole genome sequencing cannot determine since it is a reflection of cell populations and not individual cells. It is recognized that the authors' previous publication (reference #6) attempted to resolve the definition of CIN and has added further evidence that HGSOC cells are aneuploid, but the reader would benefit from an operational definition of CIN in this manuscript, and a clearer definition of whether the techniques used to identify alterations in the genome are reflective of CIN or genomic instability. Line 52 refers to "the whole spectrum of CIN", again highlighting the need to define CIN for the reader. Previous research operationally defined and described CIN in epithelial ovarian cancer using serial ascites samples from epithelial ovarian cancer patients: Penner-Goeke et al. The temporal dynamics of chromosome instability in ovarian cancer cell lines and primary patient samples. PLOS Genetics 13(4):e1006707, 2017. doi: 10.1371/journal.pgen.1006707; and, Morden et al. Chromosome instability is prevalent and dynamic in high-grade serous ovarian cancer patient samples. Gynecologic Oncology 161(3): 769-778, 2021. doi: 10.1016/j.ygyno.2021.02.038. Identifying whether the manuscript is examining genomic instability rather than CIN will help reduce the confusion in the literature brought about by individuals using the terms interchangeably.

Lines 24-25, "HGSOC is a heterogeneous, chromosomally unstable cancer with predominant somatic copy number alterations (SCNAs) and other structural variants including large-scale chromosomal rearrangements." It would benefit the reader if references were included to support this statement and provide them with background evidence.

Line 29, the references need to be separated by a comma ("56" change to "5,6").

Line 33, "remove "are" from "…are generally are molecularly…"

Lines 41-43 "Despite the relatively high prevalence of HRD and BRCA1/2 mutations in the clinic, there are only very few relevant models suggesting selection against cell lines and PDO that carry BRCA1 and BRCA2 deleterious mutations." This sentence might read better if it were split into two sentences, e.g., "Despite the relatively high prevalence of HRD and BRCA1/2 mutations in the clinic, there are only very few relevant models. This suggests selection against cell lines and PDOs that carry BRCA1 and BRCA2 deleterious mutations."

Lines 53-53, "PDOs showed…transcriptional heterogeneity between models." Is "models" supposed to represent data obtained from PDOs from different patients, PDOs from the same patient, or some other model system? It seems that "samples" be a more accurate descriptor than models.

Lines 56-57 "The PDO models we present thus shed light on evolutionary characteristics of HGSOC …". It is unclear how these models provide information about the evolutionary characteristics of HGSOC unless they were taken from the same patient at different time points in their care.

Lines 62-64, "We tested the effect of two published media compositions on 15 independent cultures and found similar PDO viability (Figure 1b)." Figure 1b shows copy number signatures and does not indicate viability. Also, the legend states "b Stacked bar plots show copy number signature activities for organoids and the matched ascites tissue sample from which they were derived"; ascites is not a tissue, and therefore more accurate language to describe cells isolated from ascites must be used.

Line 65 "The efficiency of establishing PDOs was dependent on the type of tissue sample used for derivation…". This data should be shown in the main article or as Supplemental information.

Line 68, "…serially passaged >5 times…"; it would be helpful to understand if any of the genomic or drug sensitivity data were obtained from later passage PDOs or if this was done using early passage PDOs alone. This would help clarify how long the parental tumour phenotype was maintained in serially passaged PDOs. Indeed, unpublished reports from the epithelial ovarian cancer research community working on 3D tumoursphere and PDO development indicate selection towards cellular homogeneity is typically observed at later passages. It is critical to demonstrate whether the histotype and major mutations or IHC markers of the original tumour are maintained at later passages.

Line 75 "Supplementary Figure 3b" was not included in the manuscript; only 3a is shown. Also, "value" is not defined.

Line 76, Line 75 "Supplementary Table 4" was not included in the manuscript.

Lines 86-88, S4 is not defined.

The authors mention on line 108 that signature s5 is not well represented. While this signature is represented in a number of samples present in public tumour datasets (Figure 1c), the authors state on line 109 that this is consistent with previous observations, but do not provide a reference for this. It would be helpful for the authors to expand on why the signature may be rare in the tumourspheres included in this study.

The authors mention that the tumourspheres represent good models to study the different CNV signatures. It would be helpful to note what proportion of a signature is required for the organoid to be considered a good model to study a particular signature.

Line 127, Figure 1d was not included in the manuscript.

Supplementary Table 1. It is not clear which patient samples were used to establish the PDOs. If all of these were used to establish a PDO, this can be clarified in the Table title or legend. To what does "Line" refer? Were PDOs created from each line? If not, which line was used to generate the successful PDO that is reported in the manuscript? Having treatment dates listed in the table is a potential identifier (they would not be acceptable to our local human ethics board). These should be removed and the number of months or weeks should be considered as an acceptable alternative. The term "End of treatment" is truncated in the table.

Figure 2b, PCA is used to cluster samples based on the similarity of gene expression profiles. There is no information on the methods relating to how these analyses were performed. Do the samples that clustered together based on gene expression also have similar CNV signatures? This information should also be provided for Figure 2e.

Figure 2d, A rationale for why the three organoids with the lowest gene expression and CN were selected as the comparator group should be provided.

Line 179 "We compared drug sensitivity between PDOs and their parental uncultured patient-ascites." It is not clear what this means. Was drug sensitivity compared to the patient chart data showing response to chemotherapy, or do the authors have a means of testing drug sensitivity in uncultured cells present in ascites? Please clarify.

Lines 179-181 "Using 13 anti-cancer compounds… we found moderate to high correlation between the plasma drug concentration-time…"; what are these compounds? Define "plasma drug concentration-time"; this infers that there is data on drug pharmacokinetics from these patients (plasma).

Lines 182-184, specifically "we observed no effect with the targeted therapies"; which targeted therapies – this information is not included in the manuscript. Overall, the section on PDO drug screening is poorly described and not well integrated into the manuscript.

Figure 3c. The term "Normalized GSEA Score" is truncated in the figure; it is not clear if the "padj" scale is also truncated.

For the single-cell whole genome sequencing analyses, the authors should include information relating to how they selected which organoids they would include in these analyses.

Supplementary Table 2 does not add any useful information to the manuscript and the information can simply be stated in the text.

Supplementary Table 3. The organization of this Table is unusual and it would be easier to review if the PDO numbers were shown in numerical progression (e.g., starting with #1) rather than randomly. The term "Sample name" at the top of the column is truncated. Sample type, A or O, must be defined in the Table legend. The terms "cDNA effect" and "Protein effect" are unusual. More common terms such as "cDNA modification" or "protein modification" should be considered, and an indication of what type of consequence the change produces (e.g., inactivating mutation, variant of unknown significance, etc.) should be added.

Additional comments related to Methods:

For bulk shallow whole-genome sequencing (lines 317-323), it would be helpful to mention the sequencing coverage.

It is unclear why single-end sequencing was used for the bulk whole genome sequencing (line 321), while paired-end sequencing was used for the single cell whole genome sequencing (line 345). This must be clarified.

Absolute copy number signature analysis (lines 324-325) can be combined with the previous section as this section is only one sentence long.

TCGA, Britroc, and PCWAG (line 326) – this title could be more informative and provide information relating to the analyses performed.

For single-cell shallow whole genome sequencing (lines 339-345), the quality control steps that were performed prior to CNV calling should be indicated.

Line 377 – justification for why GRCh38 was used here, but the remaining analyses use GRCh37 must be provided.

---

## [Author Response]

Reviewer #1 (Recommendations for the authors):1. The basis for categorizing CIN into S1 through S7 needs to be explained with better clarity.

Thank you for highlighting this. We have made modifications to the introduction to clarify our approach for deriving copy number signatures.

2. Page#1: The point that "recurrent somatic substitutions are rare and involve <10 driver genes", if these mutations involve mutator genes, this is likely to involve many additional genes. For instance, how would mutations in MSH2, or p53 contribute to additional mutations?

This statement refers specifically to *recurrent* somatic substitutions that have been described in high-grade serous ovarian carcinoma. The purpose of this sentence is to explain that only a relatively few genes are recognized as common drivers. We provide references to support this assertion and explain in the introduction that driver *TP53* mutations may be permissive for diverse causes of CIN.

3. Page#3: While it is interesting to note that mutations were found in BRCA1/2, were mutations also detected across passages of the organoids in p53 and BRCA1/2?

The aim of our experiments was to demonstrate efficiency and relevance of PDO to the understanding of clinically relevant mechanisms of chromosomal instability. We have not conducted longitudinal studies in these models to address the question you raise.

4. Page#3, Line 79-81: Remarkably, all 8 PDOs resulted in solid implants. Did these PDX also recapitulate mutations as found in the PDOs?

The aim of these experiments was to demonstrate that PDO are excellent tools for developing PDX models. Our work shows preservation of driver mutations between donor patients and derived PDOs. We have not conducted longitudinal evolutionary studies in the PDXs.

5. Page#4: From the spectrum of mutations highlighted, it seems obvious that the PDOs show conserved mutations, while PDOs also show considerable heterogeneity. Would this finding then contradict the premise that PDOs clonally evolve and may not necessarily recapitulate mutations inherent in the HSCOGs?

Figure 1 on page 4 shows that the PDO we have generated nearly fully represent the considerable heterogeneity between patients and have stable copy number signatures as compared to donor tissue. We have not examined clonal evolution in PDO but we clearly show that PDO retain driver mutations and copy number signatures from the donor and contain different clonal populations.

6. Page#7: Do most PDOs show a ploidy of 2 and 3.5? Does the ploidy increase correlate with an increase in the diverse subtypes of CIN?

Diploid samples appear to show fewer active signatures compared to samples of higher ploidy (2.5-3), but these findings don't reach statistical significance (linear regression of the fraction of inactive signatures as a function of ploidy, coef=-0.09, p-value=0.169). PDO1 is an example of a high-ploidy PDO with two inactive signatures.

**Author response image 1. sa2fig1:** Correlation between ploidy and the number of inactive signatures in PDOs.

7. From this data, it is unclear how significant these changes/alterations in CIN are.

We are unsure what this assertion refers to. Please could you clarify which data you are referring to and what is meant by “significant”. Do you use “CIN” to refer to copy number signature exposures? We had shown in the barplots of Figure 1 that PDO retain copy number signatures from the donor. We have now added new results showing that there is no statistical differential abundance of copy number signatures between ascites and PDOs, by use of a multivariate compositional model.

8. Figure#1: What is the extent of heterogeneity across the PDOs? this is not evident from this data.

Figure 1c (now Figure 1d in the updated manuscript) shows how the three major clades observed in patient cancer samples from clustering of their copy number signature exposures are well represented by our PDO patient-derived organoids. The clustering shows significant heterogeneity within the three major clades.

9. Figure#2: Transcriptomic analyses of the PDOs reveals critical aspects of how expression profiles of HSCOGs correlate with their transcriptome. Is there a mechanistic basis for the enrichment of the HR against the NHEJ pathway in certain PDOs?

The mechanistic basis for the low HR scores (indicating HR deficiency) is explained by mutations in genes in the HR pathway. For example, PDO8, which has the lowest HR score, contains a clinically pathogenic mutation in *BRCA1* (Figure 1—figure supplement 3). Although the KEGG ssGSEA scores shown in Figure 2e (now Figure 3e) are useful to compare samples to each other, scores from different pathways are not directly comparable, and one must be careful when attributing a functional category to a sample based solely on enrichment scores.

10. Figure#3: The PDO drug screening is an interesting and revealing experiment. However, the explanation is rather confusing, since the demarcation between sensitive and resistant PDOs does not include all the PDOs under each category. Figure 3A: Could include representatives from sensitive and resistant categories with a label accordingly for each of the two categories. Figure 3C: Is this data derived from each PDO from the sensitive and resistant categories? or is this pooled data? this is not clear from the Results section. Although the GSEA enrichment analyses enrich for MYC targets, E2F and G2/M checkpoint targets.

The purpose of the line fitting of response data in Figure 3a (now Figure 4a) is to show that there is a very high correlation for drug response between uncultured donor cells from the patient and the derived PDOs for all treatments. This data establishes that PDO retain appropriate sensitivity and resistance. In Figure 3b (now Figure 4b) we have now indicated sensitive PDOs with a blue dot and resistant PDOs with a red dot and explained in the text that the data shown only includes organoids with successful RNA-Seq data. Figure 3c (now Figure 4c) is derived from the comparison between sensitive and resistant PDOs as pooled data. The figure caption has been rewritten to improve clarity.

11. Figure#4: This figure is not numbered. It would be useful to indicate and explain each panel. Label the left panels as A-C and insert the legend on the X and Y-axis respectively. Likewise, label the columns on the right and label as D-F.

Thank you. These suggestions have been implemented and this is now Figure 5.

Reviewer #2 (Recommendations for the authors):1. The authors performed single-cell whole genome sequencing on three PDOs (PDO2, PDO3, PDO6). It is not clear why the authors chose these three PDOs for single-cell whole-genome sequencing. This method is good for organoid intratumoural heterogeneity study if the authors provide more samples of single-cell whole genome sequencing.

For the single-cell shallow whole genome sequencing three organoids were selected arbitrarily as representative models and we included two fast-growing PDOs (PDO2 and PDO3) and one slow-growing model (PDO6).

2. In line 147, the authors said PDOs consist of 100% tumour cells, are you sure?

As shown in our targeted-gene mutation analysis, all our PDOs contain a *TP53* allelic fraction between 80-95% and this information is now included in the text for clarification. Moreover, in our single cell DNAseq data we do not observe any normal copy number profiles that would indicate normal cells.

3. In the PDO drug screening part, the authors performed gene expression and pathway analysis on sensitive and resistant groups. How about the mutations and CNAs?

As HGSOC is predominantly driven by CNA not by somatic substitutions, we investigate gene expression driven by CNA in Figure 3c and 3d. We also present whole genome correlations between copy number and expression in Figure 2—figure supplement 1, along with CNA data in Figure 2 together with Figure 1—figure supplement 6. Our mutational analysis is described in Figure 1—figure supplement 3. We compared PDO sensitive and resistant groups for ploidy and number of copy number segments and did not observe any significant differences (the average number of segments is 167 for sensitive and 183 resistant PDOs, and the average ploidies are 2.8 for sensitive and 2.46 for resistant PDOs; p-value = 0.7441 and p-value = 0.2374 respectively; Welch Two Sample t-test). These data have been added to the manuscript.

Reviewer #3 (Recommendations for the authors):This was very challenging to read due to lack of information (e.g., Figure 1d, Supplementary Figure 3b, and Supplementary Table 4 were not included with the manuscript), and limited or lacking justification for the inclusion of data. Some of the figures do not adequately transmit the intended information and the figure legends should be revised to provide more comprehensive information to the reader. Justification for some analytical approaches (e.g., selection of gene expression and CN comparison groups) would enhance the manuscript. Please see specific comments below.Organoids are typically composed of numerous cell types to recapitulate an organ, whereas spheroids are typically constituted of a single cell type grown in a 3-dimensional format. A weakness of the manuscript is the lack of evidence demonstrating the composition of the PDOs and the lack of an operational definition of an organoid. Without further evidence, these 3D cultures should be referred to as tumourspheres or spheroids. Some of the most informative research on the development of multicellular models to recapitulate high-grade serous ovarian cancer are from the Balkwill laboratory (see Malacrida at al. Building in vitro 3D multicellular models of high-grade serous ovarian cancer. STAR Protocols 3(1): 101086. DOI: 10.1016/j.xpro.2021.101086). In general, the epithelial ovarian cancer field accepts the definition of organoids as those with multiple cell types (e.g., tumour cells, mesothelial cells, and lymphocytes).

We have addressed this comment above in the public review section. In the Malacrida MS (and associated methods you cite), HGSOC cell lines were cocultured with other cells on specific matrices.

CIN is a type of genomic instability that is broadly defined as an increased rate of chromosome gains or losses and is best identified through analysis of single cells (e.g., karyotype analysis), something that bulk whole genome sequencing cannot determine since it is a reflection of cell populations and not individual cells. It is recognized that the authors' previous publication (reference #6) attempted to resolve the definition of CIN and has added further evidence that HGSOC cells are aneuploid, but the reader would benefit from an operational definition of CIN in this manuscript, and a clearer definition of whether the techniques used to identify alterations in the genome are reflective of CIN or genomic instability. Line 52 refers to "the whole spectrum of CIN", again highlighting the need to define CIN for the reader. Previous research operationally defined and described CIN in epithelial ovarian cancer using serial ascites samples from epithelial ovarian cancer patients: Penner-Goeke et al. The temporal dynamics of chromosome instability in ovarian cancer cell lines and primary patient samples. PLOS Genetics 13(4):e1006707, 2017. doi: 10.1371/journal.pgen.1006707; and, Morden et al. Chromosome instability is prevalent and dynamic in high-grade serous ovarian cancer patient samples. Gynecologic Oncology 161(3): 769-778, 2021. doi: 10.1016/j.ygyno.2021.02.038. Identifying whether the manuscript is examining genomic instability rather than CIN will help reduce the confusion in the literature brought about by individuals using the terms interchangeably.

Thank you. We addressed this comment in the public review section and have included text to clarify these issues.

Lines 24-25, "HGSOC is a heterogeneous, chromosomally unstable cancer with predominant somatic copy number alterations (SCNAs) and other structural variants including large-scale chromosomal rearrangements." It would benefit the reader if references were included to support this statement and provide them with background evidence.

Thank you. References supporting this assertion have been added.

Line 29, the references need to be separated by a comma ("56" change to "5,6").

Thank you, this has been changed.

Line 33, "remove "are" from "…are generally are molecularly…"

Amended.

Lines 41-43 "Despite the relatively high prevalence of HRD and BRCA1/2 mutations in the clinic, there are only very few relevant models suggesting selection against cell lines and PDO that carry BRCA1 and BRCA2 deleterious mutations." This sentence might read better if it were split into two sentences, e.g., "Despite the relatively high prevalence of HRD and BRCA1/2 mutations in the clinic, there are only very few relevant models. This suggests selection against cell lines and PDOs that carry BRCA1 and BRCA2 deleterious mutations."

Thank you. We have improved the wording.

Lines 53-53, "PDOs showed…transcriptional heterogeneity between models." Is "models" supposed to represent data obtained from PDOs from different patients, PDOs from the same patient, or some other model system? It seems that "samples" be a more accurate descriptor than models.

We have changed models to PDO as we are not referring to the patient samples.

Lines 56-57 "The PDO models we present thus shed light on evolutionary characteristics of HGSOC …". It is unclear how these models provide information about the evolutionary characteristics of HGSOC unless they were taken from the same patient at different time points in their care.

Even though three pairs of organoids were taken from the same patients, the single cell analysis is only performed on singleton samples. Therefore, we have reworded this sentence to “The PDO models we present thus shed light on the diversity of chromosomal landscapes of HGSOC”.

Lines 62-64, "We tested the effect of two published media compositions on 15 independent cultures and found similar PDO viability (Figure 1b)." Figure 1b shows copy number signatures and does not indicate viability. Also, the legend states "b Stacked bar plots show copy number signature activities for organoids and the matched ascites tissue sample from which they were derived"; ascites is not a tissue, and therefore more accurate language to describe cells isolated from ascites must be used.

We apologise for the mis-referencing, as it is Supplementary Figure 1b (now Figure 1—figure supplement 1A) which indicates viability. Regarding the terminology used in the caption, we had adopted the use of the word "tissue" to refer to ascites following the Human Tissue Act definition of 'relevant material' as any material from a human body that consists of, or includes, cells. However, following the reviewer’s comment we have changed this to simply "ascites sample".

Line 65 "The efficiency of establishing PDOs was dependent on the type of tissue sample used for derivation…". This data should be shown in the main article or as Supplemental information.

This has been changed as suggested.

Line 68, "…serially passaged >5 times…"; it would be helpful to understand if any of the genomic or drug sensitivity data were obtained from later passage PDOs or if this was done using early passage PDOs alone. This would help clarify how long the parental tumour phenotype was maintained in serially passaged PDOs. Indeed, unpublished reports from the epithelial ovarian cancer research community working on 3D tumoursphere and PDO development indicate selection towards cellular homogeneity is typically observed at later passages. It is critical to demonstrate whether the histotype and major mutations or IHC markers of the original tumour are maintained at later passages.

All data generated in this paper was done with PDOs at passages 5–15. Clarification has been added to the main text.

Line 75 "Supplementary Figure 3b" was not included in the manuscript; only 3a is shown. Also, "value" is not defined.

We apologise for this – only the first subfigure of Supplementary Figure 3 (now Figure 1—figure supplement 3) had been labelled. “value” has been replaced by “patient”.

Line 76, Line 75 "Supplementary Table 4" was not included in the manuscript.

We apologise for this omission and it is now included.

Lines 86-88, S4 is not defined.

S4 has now been defined in the main text.

The authors mention on line 108 that signature s5 is not well represented. While this signature is represented in a number of samples present in public tumour datasets (Figure 1c), the authors state on line 109 that this is consistent with previous observations, but do not provide a reference for this. It would be helpful for the authors to expand on why the signature may be rare in the tumourspheres included in this study.

In line 108 we point out that the general abundance of s5 is low in organoids, as it is in the general mutational spectrum of samples, but not that it is underrepresented. We mention that its general low abundance is due to chromothripsis being a rare event, for which we have added citations.

The authors mention that the tumourspheres represent good models to study the different CNV signatures. It would be helpful to note what proportion of a signature is required for the organoid to be considered a good model to study a particular signature.

We would be more cautious about what these data indicate to date: the organoids presented successfully represent primary tissue when characterized according to CNV signatures, and the availability and characterization of organoids can help us decipher further the biological mechanisms behind each signature. Finding the precise thresholds to classify tumours according to the presence of mutational processes is still ongoing work. In unpublished results, and in the case of WGD, s4 (especially in combination with s3) show excellent performance in classifying tumours between WGD and non-WGD using a support vector machine. Similar analyses for the remaining signatures are pending, but we advise against finding threshold using proportions of signatures, as these data are compositional.

Line 127, Figure 1d was not included in the manuscript.

We apologise for the typo – Line 127 should have referred to Figure 1c, but now correctly refers to Figure 1d.

Supplementary Table 1. It is not clear which patient samples were used to establish the PDOs. If all of these were used to establish a PDO, this can be clarified in the Table title or legend. To what does "Line" refer? Were PDOs created from each line? If not, which line was used to generate the successful PDO that is reported in the manuscript? Having treatment dates listed in the table is a potential identifier (they would not be acceptable to our local human ethics board). These should be removed and the number of months or weeks should be considered as an acceptable alternative. The term "End of treatment" is truncated in the table.

Tables have now been changed as suggested.

Figure 2b, PCA is used to cluster samples based on the similarity of gene expression profiles. There is no information on the methods relating to how these analyses were performed. Do the samples that clustered together based on gene expression also have similar CNV signatures? This information should also be provided for Figure 2e.

The PCA in Figure 2b (now Figure 3b) was created using all genes, normalised using the DESeq2 method based on gene-specific geometric means – this has now been added to the Methods section. These first two principal components are not obviously related to CN aberrations. PDO2 and PDO10, which appear together in the PCA, have markedly different CNV signatures, and no other trend regarding signatures is observed.

**Author response image 2. sa2fig2:** Correlation of number of segments and ploidy with PC1 and PC2 values.

Figure 2d, A rationale for why the three organoids with the lowest gene expression and CN were selected as the comparator group should be provided.

The three organoids, out of eleven, with the lowest expression have been selected with the idea that it is a number high enough not to include solely outliers, but that it leaves out a high enough number of organoid samples (eight) in which we can observe the variability in their copy number and gene expression. There is a very high correlation between these averaged GE values when using the lowest three organoids, and when using the lowest two, or four.

**Author response image 3. sa2fig3:** Comparing average gene expression values when summarising the lowest two or three organoids (left) and three or four (right).

Line 179 "We compared drug sensitivity between PDOs and their parental uncultured patient-ascites." It is not clear what this means. Was drug sensitivity compared to the patient chart data showing response to chemotherapy, or do the authors have a means of testing drug sensitivity in uncultured cells present in ascites? Please clarify.

We performed in vitro drug testing using uncultured primary patient ascites cells using standard of care drugs and targeted therapies. The drugs used are now listed in the Methods section.

Lines 179-181 "Using 13 anti-cancer compounds… we found moderate to high correlation between the plasma drug concentration-time…"; what are these compounds? Define "plasma drug concentration-time"; this infers that there is data on drug pharmacokinetics from these patients (plasma).

All the compounds are now stated in the Methods section. Apologies for the typo but “plasma” was an error and has been removed from the text.

Lines 182-184, specifically "we observed no effect with the targeted therapies"; which targeted therapies – this information is not included in the manuscript. Overall, the section on PDO drug screening is poorly described and not well integrated into the manuscript.

We tested uncultured ascites cells from the donor patient and compared this to the response of the patient-derived organoids from which they were derived. We have expanded on and clarified this section.

Figure 3c. The term "Normalized GSEA Score" is truncated in the figure; it is not clear if the "padj" scale is also truncated.

This figure is no longer cropped.

For the single-cell whole genome sequencing analyses, the authors should include information relating to how they selected which organoids they would include in these analyses.

As stated in Reviewer 2 Point 1, three organoids were selected at random to ensure an unbiased analysis. Those we were two PDOs that grew fast (PDO2 and PDO3) and one slow growing model (PDO6).

Supplementary Table 2 does not add any useful information to the manuscript and the information can simply be stated in the text.

The information has been added to the main text and Supplementary Table 2 has been deleted.

Supplementary Table 3. The organization of this Table is unusual and it would be easier to review if the PDO numbers were shown in numerical progression (e.g., starting with #1) rather than randomly. The term "Sample name" at the top of the column is truncated. Sample type, A or O, must be defined in the Table legend. The terms "cDNA effect" and "Protein effect" are unusual. More common terms such as "cDNA modification" or "protein modification" should be considered, and an indication of what type of consequence the change produces (e.g., inactivating mutation, variant of unknown significance, etc.) should be added.

Table has been changed according to the reviewer suggestions (now Supplementary File 2).

Additional comments related to Methods:For bulk shallow whole-genome sequencing (lines 317-323), it would be helpful to mention the sequencing coverage.

Our shallow whole-genome samples have an approximate coverage of 0.25-0.3, assuming a diploid genome (an assumption which is often broken). This has been added to the manuscript.

It is unclear why single-end sequencing was used for the bulk whole genome sequencing (line 321), while paired-end sequencing was used for the single cell whole genome sequencing (line 345). This must be clarified.

The analysis pipeline to call absolute copy number and then signatures is based on the R package QDNAseq, which requires single read data, and this is the reason behind using single-end sequencing in shallow whole-genome sequencing data.

The library for single cells was made using 10X CNV and we used their recommendation for sequencing to use paired-end sequencing of 150bp per read – the pipeline for copy number calling is completely different from that above in which QDNAseq is used.

Absolute copy number signature analysis (lines 324-325) can be combined with the previous section as this section is only one sentence long.

This change has been made.

TCGA, Britroc, and PCWAG (line 326) – this title could be more informative and provide information relating to the analyses performed.

The title has been changed to “Comparison of organoid copy number signatures to those of TCGA, BriTROC-1 and PCAWG”.

For single-cell shallow whole genome sequencing (lines 339-345), the quality control steps that were performed prior to CNV calling should be indicated.

We have added the following sentence: “The Cell Ranger pipeline was used for quality control, trimming and alignment.”

Line 377 – justification for why GRCh38 was used here, but the remaining analyses use GRCh37 must be provided.

GRCh37 was used for bulk sWGS signature analysis following the pipeline from Macintyre et al. At the used resolution of 30 kb the difference in using GRCh37 or GRCh38 is minimal. However, in the gene-specific analyses such as RNA-Seq, in which changes in the *gtf* files might be more important, the most recent version GRCh38 was used.